# Patient and guardian perspectives on tissue engineering in microtia reconstruction

Arya Sherafat[1], James Antongiovanni[2,3], Chizoba Mosieri[2,4], Asthon Rosenbloom[2,5], Lana Mamoun[2,6], Sierra Willens[2], Leo Alaniz[2], Rahim Esfandyarpour[7], Mary Ziegler[2], Miles J. Pfaff [2,8]*

1 University of California Riverside, School of Medicine, Riverside, California, United States of America, 2 Department of Plastic Surgery, University of California Irvine Medical Center, Orange, California, United States of America, 3 Washington State University Elson S. Floyd College of Medicine, Spokane, Washington, United States of America, 4 Louisiana State University Health Shreveport, School of Medicine, Shreveport, Louisiana, United States of America, 5 American Medical Program at Tel Aviv University Tel Aviv, Illinois, United States of America, 6 California University of Science and Medicine, Colton, California, United States of America, 7 University of California Irvine, School of Engineering, Irvine, California, United States of America, 8 Plastic Surgery, Children's Hospital of Orange County, Orange, California, United States of America

* mlies.pfaff@gmail.com

## Abstract

### Background

Tissue Engineering (TE) is a newer technology with ongoing development across various areas of healthcare. For microtia patients, TE holds promise as a viable option for ear reconstruction, making it essential to understand the perspectives of patients and guardians. This study aimed to evaluate the perspectives of microtia patients and guardians, specifically investigating levels of awareness, comprehension, and interest in TE.

### Methods

A survey and educational material were distributed to microtia patients and guardians. Surveyors were assessed on a Likert scale, and comparative analyses, including two-sample t-testing and multivariate analysis with logistic regression, were used to observe trends in responses.

### Results

A total of 40 surveys were recorded. White patients reported greater familiarity with TE. More than half of the respondents expressed an interest in TE as a reconstructive option. Having a doctor provide TE information positively correlated with willingness to adopt TE reconstruction. Understanding of TE positively correlated with willingness to pursue TE for reconstruction. Patients with more severe grade microtia and white patients were more likely to travel to a hospital that offered TE.

**Data availability statement:** The de-identified patient data has previously been submitted alongside the paper. As for a non-author contact, we identify Van Hoang, who is the clinical research coordinator at CHOC (Children's Hospital Orange County), where our research was conducted (Van Hoang). Additionally, the REDCap administrator for our survey study at CHOC is Kevin Bostwick (Kevin.Bostwick1@choc.org).

**Funding:** The author(s) received no specific funding for this work.

**Competing interests:** The authors have declared that no competing interests exist.

## Conclusions

We found that TE is an unfamiliar concept for most patients. Demographic, socio-economic, cost, and trust in a doctor played an essential role in patients' and guardians' willingness to choose TE for microtia reconstruction. As TE for reconstruction approaches clinical use, these key considerations from a patient and family-centered lens about TE-based reconstruction is critical for application.

## Introduction

Tissue engineering (TE) represents a frontier in modern medicine with the potential to revolutionize treatment options across various clinical disciplines. In the realm of reconstructive surgery, microtia—a congenital deformity characterized by underdeveloped external ears—holds significant transformative potential for TE [1]. A survey administered to the American Society of Plastic Surgeons indicated that 59% of respondents believe that TE will represent the gold standard in microtia reconstruction within 15 years [2]. Historically, ear reconstruction, often using autologous rib cartilage or synthetic materials, has been the standard for microtia treatment. While these procedures have their benefits, they each come with inherent limitations [3–6].

TE offers the prospect of creating patient-specific, biocompatible tissues for reconstructive purposes. Advancements in stem cell research, scaffold development, and bio-fabrication have furthered the feasibility of TE in clinical settings. For example, mesenchymal stem cells have demonstrated remarkable potential for differentiating into chondrocytes [7]. Scaffold development has progressed to complex 3D-printed structures that mimic the extracellular matrix, providing support for cell growth and tissue formation [8]. Advancements such as these are further bridging the gap between laboratory research and clinical application.

Preliminary research has demonstrated the successful application of TE techniques utilizing autologous cell-engineered chondrocytes for total ear reconstruction in cases of microtia [9]. Integration of such advancements into clinical practice requires an understanding beyond technical capabilities. The acceptance, reservations, and perspectives of patients and their families play a crucial role in determining the future trajectory of these interventions. Their concerns about the safety and efficacy of TE-based approaches, as well as their understanding of its long-term implications, play a critical role in decision-making processes. While studies have reviewed the scientific, clinical, and ethical implications of TE [10], the perspectives of patients and their families are limited. Understanding the concerns, beliefs, and expectations of families is essential for aligning TE innovations with patient needs and preferences.

This study aimed to explore patient and family perspectives on TE. Utilizing the concept of TE for microtia reconstruction, this study identified key concerns and priorities from a patient and family-centered lens about TE-based reconstruction. By capturing these insights, this study contributes to a foundational, patient-centered framework for the development and application of TE that extends beyond microtia

reconstruction. Ultimately, this approach ensures that TE-driven innovations are not only technologically advanced, but also aligned with the values and priorities of the patients they are designed to treat.

## Materials and methods

### Participants

Microtia patients at the Children's Hospital of Orange County (CHOC) craniofacial clinic were identified and recruited by chart review between August 15th 2024 and March 27th 2025. Eligible patients were those aged 0–21 years with a diagnosis of Microtia (ICD Q17.7), who had not had microtia reconstruction or were at least one year postoperative from their most recent ear reconstructive procedure. A survey was conducted to assess overall awareness and perceptions of TE in microtia reconstruction, covering demographics, interest, concerns, cultural implications, cost, considerations, and travel factors. It was administered to eligible patients or their guardians. Patients aged 16 years or older completed the survey themselves, while guardians completed the survey on behalf of patients younger than 16. Verbal informed consent was obtained from patients and guardians by clinical staff and securely recorded in RedCap survey management software. All surveys were carried out in accordance with the Institutional Review Board (IRB) at CHOC (IRB #240218).

### Educational materials

An educational document was distributed to families prior to the initiation of the survey. Educational materials were provided in either English or Spanish, aimed at introducing the potential application of TE as a surgical approach for microtia reconstruction. Key considerations presented within the document included the utilization of the patient's own native tissue, the potential for reducing the number of surgical interventions, cost implications, safety profile, research developments, and accessibility. Current reconstruction options for microtia were also outlined. These topics were intentionally framed in an unbiased manner to facilitate the development of an informed perspective among patients previously unfamiliar with TE. Following the distribution of the educational document, the patient or guardian answered the perspectives survey.

### Survey

Two surveys were designed to evaluate perspectives and insights on TE. The first targeted microtia patients aged 16 years and older, while the second was adapted to assess the views of guardians of microtia patients who met the inclusion criteria. The surveys collected demographic information, including patient date of birth, gender, race, and ethnicity. Participants were then presented with a series of 20 questions evaluated using a five-point Likert scale.

### Data analysis & grouping

Qualitative data collected via Likert scales were converted to a numerical scoring system and securely entered into Microsoft Excel (Version 16.98) for analysis. Survey responses were stratified based on demographic characteristics, microtia-reconstruction status, perceived barriers, and levels of hesitation or trust. Survey questions were categorized into 4 domains: (1) understanding of microtia/TE, (2) willingness to adopt TE, (3) trust in TE, and (4) barriers to accessing TE. Groupings for barrier-related and hesitation/trust-related differences were derived from responses to specific survey items. Responses scoring 4 or 5 on the Likert scale—indicating agreement or support—were categorized as one group, while scores of 1 or 2—indicating disagreement or disapproval—comprised the second group. Responses with a score of 3 (neutral) and those marked as 'prefer not to answer' were excluded from group comparisons. Comparative analyses between groups were conducted using two-sample t-tests. For each survey item, mean Likert scores and one-tailed p-values were calculated. Non-parametric correlation using Spearman's rank correlation was used to assess the relationship between variables and survey domains. Multivariate analysis, using ordinal logistic regression, was employed to

examine the relationship between demographic and operative variables and survey responses. Lastly, health record-derived patient insurance plans and zip-code data were used to assess the Childhood Opportunity Index (COI) [11], which evaluates the quality of resources and neighborhood conditions. The indicators in zip-code level differences for COI calculations include childhood education quality, pollution, healthy environments, safety- and health-related resources, economic opportunities and resources, housing resources, social resources, wealth, and more. z-scores calculating these indicators were used to categorize at or below the 20th percentile as "very low" COI. Tracts above the 20th and at or below the 40th percentile were classified as "low," above the 40th percentile and at or below the 60th percentile were categorized as "moderate," above the 60th percentile and at or below the 80th percentile were categorized as "high," and lastly tracts above the 80th percentile were classified as "very high" opportunity. COI and insurance type data were used as proxies to determine socioeconomic status and access to resources. An unadjusted ordinal logistic regression analysis was conducted using COI data, insurance status, and individual questions. A power analysis was conducted to determine the required sample size. An effect size of 1.16 was utilized [12], demonstrating a minimum sample size of 13 surveyors per group compared (power = 0.08, alpha = 0.05). All statistical analyses were conducted in either IBM SPSS Statistics, Version 28.0.1 (IBM Corp., Armonk, NY) or RStudio, Version 2024.09.0 + 375 (RStudio, PBC, Boston, MA). All survey data was accessed and used for analysis from April 1st 2025 to June 21st 2025 with the absence of patient identifying information.

## Results

### Surveyor demographic factors

A total of 40 surveys were collected, comprising 30 guardian and 8 patient surveys. Two surveys were collected from guardians on behalf of patients aged 16 and above. The majority of surveyors were female (n = 32, 80%) and identified as Hispanic or Latino (n = 31, 78%) (Table 1a). Fifteen patients had at least one microtia reconstructive procedure, while 25 had no prior reconstruction. Over a third of patients (n = 15, 38%) had grade 3 microtia, and most had unilateral involvement (n = 28, 70%) (Table 1b). Most surveyors indicated first learning of TE information from medical staff (n = 27, 68%) (Table 1c). Many patients were from lower COI areas (n = 27, 55%), indicating limited access to resources for surveyor populations.

### Quantitative survey evaluation

Many participants were unfamiliar with TE indicating that they did not know 'what tissue engineering is before today' (n = 29/40, 72.5%). Surveyors stated that they had learned about it via medical staff (n = 27, 68%), followed by their doctor (n = 10, 25%), and the internet (n = 2, 5%). Following their review of the educational material, half of respondents (n = 20/40, 50%) indicated that they would 'choose tissue engineering over other ways to fix the ear' and more than half (n = 26/40, 65%) indicated interest in TE for microtia reconstruction over traditional methods.

Most surveyors (n = 32/40, 80%) believed TE is a good idea for medicine/surgery. Nearly all the participants (n = 38/40, 95%) indicated they would want to see 'pictures of other kids' results before choosing tissue engineering for themselves or their child. The cost of TE played a role in more than half of the surveyors' (n = 22/40, 55%) decisions to choose TE for their microtia reconstruction.

### Demographic differences *in* survey results

Guardians were more familiar with the concept of TE than patients (2.44 vs. 1.63 and 2.50 vs. 1.63, p < 0.05 and p = 0.01, respectively). Patients wanted to see pictures of other patients' TE microtia reconstruction results before choosing TE for their reconstruction (4.50 vs. 4.13, p < 0.05). More patients indicated a cultural reason for not choosing TE compared to guardians (2.5 vs. 1.94, p < 0.05). White patients were more familiar with TE before the introduction of educational

**Table 1. Surveyor demographic factors categorized by surveyor type, gender, race, ethinicity, age, reconstruction status, microtia status and sourcing of TE information. Frequency reported and percentage identified from a total of 40 participant surveys.**

| Table 1a. Surveyor Demographics. | | |
|---|---|---|
| **Surveyor** | **Frequency (n)** | **Percent (%)** |
| Parent | 30 | 0.75% |
| Patient (16+) | 8 | 0.20% |
| Parent on behalf of 16+Patient | 2 | 0.05% |
| **Gender of Surveyor** | **Frequency (n)** | **Percent (%)** |
| Male | 8 | 0.20% |
| Female | 32 | 0.80% |
| **Race of Surveyor** | **Frequency (n)** | **Percent (%)** |
| White | 9 | 0.23% |
| Asian | 2 | 0.05% |
| More than one race | 1 | 0.03% |
| Other | 19 | 0.48% |
| Prefer not to answer | 9 | 0.23% |
| **Ethnicity of Surveyor** | **Frequency (n)** | **Percent (%)** |
| Hispanic or Latino | 31 | 0.78% |
| Not Hispanic or Latino | 8 | 0.20% |
| Prefer not to answer | 1 | 0.03% |
| **Patient Age** | **Frequency (n)** | **Percent (%)** |
| Age 0 – Age 5 | 9 | 0.23% |
| Age 6 – Age 10 | 12 | 0.30% |
| Age 11 – Age 16 | 9 | 0.23% |
| Age 16 – Age 21 | 10 | 0.25% |

Table 1a. Surveyor demographic factors categorized by surveyor type, gender, race, ethnicity and age. Frequency reported and percentage identified from a total of 40 participant surveys.

| Table 1b. Microtia Status. | | |
|---|---|---|
| **Reconstruction Status** | **Frequency (n)** | **Percent (%)** |
| No Prior Reconstruction | 25 | 0.63% |
| Prior Reconstruction | 15 | 0.38% |
| **Uni-Lateral vs. Bi-Lateral** | **Frequency (n)** | **Percent (%)** |
| Uni-Lateral | 28 | 0.70% |
| Bi-Lateral | 12 | 0.30% |
| **Microtia Grade** | **Frequency (n)** | **Percent (%)** |
| Grade 1 | 2 | 0.05% |
| Grade 2 | 7 | 0.18% |
| Grade 3 | 15 | 0.38% |
| Grade 4 | 3 | 0.08% |
| Bi-Lateral | 9 | 0.23% |
| Unidentified | 4 | 0.10% |

Table 1b. Surveyor microtia status categorized reconstruction status, uni-lateral or bi-lateral involvement, and microtia severity grade. Frequency reported and percentage identified from a total of 40 participant surveys.

| Table 1c. TE Information. | | |
|---|---|---|
| **Source** | **Frequency (n)** | **Percent (%)** |
| Medical Staff | 27 | 0.68% |
| My Doctor | 10 | 0.25% |
| Internet | 2 | 0.05% |
| Not Applicable | 1 | 0.03% |

Table 1c. Surveyor identified primary source of learning TE based information. Frequency reported and percentage identified from a total of 40 participant surveys.

materials (3.00 vs. 2.13, $p < 0.05$). Patients with unilateral microtia were more likely to choose TE over other reconstructive approaches (3.54 vs. 2.92, $p < 0.05$) compared to patients with bilateral microtia. More unilateral microtia patients reported cultural reasons for not choosing TE (2.21 vs. 1.67, $p < 0.01$). Patients with microtia grade 1 or 2 indicated cost would play a greater role in their willingness to choose TE compared to patients with microtia grade 3 or 4 (4.00 vs. 3.32, $p < 0.05$) (Table 2).

**Table 2. Demographic Differences.**

| Question | Parent (mean) | Patient (mean) | p-value |
|---|---|---|---|
| Have you heard of 'Tissue Engineering' before today? | 2.438 | 1.625 | 0.020 |
| Did you know what 'Tissue Engineering' is before today? | 2.500 | 1.625 | 0.010 |
| Do you know the different ways doctors fix microtia/anotia? | 3.469 | 2.625 | 0.010 |
| Would you want to see pictures of other kid's results before choosing Tissue Engineering for your child? | 4.125 | 4.500 | 0.033 |
| Would there be a cultural reason as to why you may not choose Tissue Engineering? | 1.938 | 2.500 | 0.012 |
| **Question** | **White (mean)** | **Non-White (mean)** | **p-value** |
| Have you heard of 'Tissue Engineering' before today? | 3.000 | 2.130 | 0.021 |
| **Question** | **Age 10+ (mean)** | **Age 10- (mean)** | **p-value** |
| Would you want to see pictures of other kid's results before choosing Tissue Engineering for your child? | 4.318 | 4.056 | 0.055 |
| Would you seek out and travel to a hospital that offers Tissue Engineering? | 3.364 | 3.778 | 0.055 |
| **Question** | **No Prior Reconstruction (mean)** | **Prior Reconstruction (mean)** | **p-value** |
| Are you interested in learning more about Tissue Engineering? | 4.080 | 3.667 | 0.034 |
| Have you looked up information about 'Microtia' or 'Anotia' on your own? | 3.560 | 2.867 | 0.028 |
| Besides surgery in general, do you have any worries about using Tissue Engineering for ear fixing? | 2.600 | 2.067 | 0.030 |
| Would the cost of Tissue Engineering for your child's ear be a big part of your decision to choose Tissue Engineering? | 3.640 | 3.067 | 0.057 |
| Would you seek out and travel to a hospital that offers Tissue Engineering? | 3.720 | 3.267 | 0.044 |
| **Question** | **Microtia Grade 1 or 2 (mean)** | **Microtia Grade 3 or 4 (mean)** | **p-value** |
| How comfortable do you feel about new technology being used in surgeries? | 4.182 | 3.640 | 0.052 |
| Do you understand how Tissue Engineering could be used to fix body parts? | 3.636 | 3.040 | 0.053 |
| Do you know the different ways doctors fix microtia/anotia? | 2.727 | 3.400 | 0.017 |
| Would you trust information that shows Tissue Engineering worked well for other children? | 4.091 | 3.800 | 0.021 |
| Would the cost of Tissue Engineering for your child's ear be a big part of your decision to choose Tissue Engineering? | 4.000 | 3.320 | 0.037 |
| **Question** | **Uni-Lateral Microtia (mean)** | **Bi-Lateral Microtia (mean)** | **p-value** |
| Would you choose Tissue Engineering over other ways to fix the ear? | 3.536 | 2.917 | 0.033 |
| Would there be a cultural reason as to why you may not choose Tissue Engineering? | 2.214 | 1.667 | 0.005 |

Significant differences in response rates to individual questions when groups are categorized by demographic differences including surveyor type, race, age, reconstruction status, microtia grade, and microtia involvement. Averaged responses are indicated on a Likert scale of 0 (not at all) to 5 (definitely). Differences in response rates were deemed significantly different at $p < 0.05$.

Patients with higher microtia grades, as well as white and non-Hispanic patients, were less likely to consider cost a barrier in deciding to choose TE (p = 0.018, p = 0.009, p = 0.042). Patients with severe grade microtia and white patients responded that they were more likely to travel to a hospital that offered TE (p = 0.029, p = 0.030). As expected, reconstruction status played a role in TE interest. Patients who had not yet undergone ear reconstruction showed a higher interest in TE (OR = 10.52, 95% CI = 1.68–65.88, p = 0.012) and a greater willingness to adopt TE solutions for reconstruction (OR = 6.45, 95% CI = 1.52–27.33, p = 0.011). Grade 3 and Grade 4 microtia patients were more likely to be aware of surgical approaches to microtia reconstruction than Grade 1 or 2 patients (OR=5.59, 95% CI = 1.29–24.23, p = 0.021).

## Barrier differences in survey results

Those who indicated that cost played a role in their willingness to choose TE had a greater understanding of TE prior to reviewing the educational material (2.50 vs. 1.50 and 2.59 vs. 1.67, p < 0.05 and p < 0.05, respectively). Yet, these surveyors would still choose TE over other reconstructive methods (3.68 vs. 2.50, p = 0.01). Those who indicated a cultural role in their willingness to choose TE were less familiar with how TE could be used in surgical reconstruction (2.00 vs. 3.3.1, p < 0.05) and were more willing to seek out and travel to a hospital that offers TE (4.50 vs. 3.49, p < 0.05). Surveyors who indicated a willingness to travel for TE were generally more comfortable with technology in surgery (3.92 vs. 3.00, p < 0.05) and the TE approach for reconstruction (3.92 vs. 3.20, p < 0.05). These surveyors also indicated an interest in learning more about TE (4.13 vs. 3.40, p < 0.05) and would believe a doctor who stated that TE would fix their or their child's ear (3.96 vs. 3.00, p < 0.01) (Table 3).

Table 3. Barrier Differences.

| Question | Indicated Cost Role in TE (mean) | Denied Cost Role in TE (mean) | p-value |
|---|---|---|---|
| Have you heard of 'Tissue Engineering' before today? | 2.500 | 1.500 | 0.018 |
| Did you know what 'Tissue Engineering' is before today? | 2.591 | 1.667 | 0.021 |
| Do you understand how microtia/anotia occurs when a baby is born? | 2.955 | 3.833 | 0.043 |
| Would you choose Tissue Engineering over other ways to fix the ear? | 3.682 | 2.500 | 0.001 |
| Question | Indicated Cultural Role in TE (mean) | Denied Cultural Role in TE (mean) | p-value |
| Do you understand how Tissue Engineering could be used to fix body parts? | 2.000 | 3.314 | 0.049 |
| Would you seek out and travel to a hospital that offers Tissue Engineering? | 4.500 | 3.486 | 0.048 |
| Question | Indicated Travel for TE (mean) | Denied Travel for TE (mean) | p-value |
| How comfortable do you feel about new technology being used in surgeries? | 3.917 | 3.000 | 0.024 |
| How do you feel about using Tissue Engineering for your child's ear reconstruction? | 3.917 | 3.200 | 0.045 |
| Do you think Tissue Engineering is a good idea for medicine/surgery? | 4.167 | 3.400 | 0.021 |
| Are you interested in learning more about Tissue Engineering? | 4.125 | 3.400 | 0.011 |
| Are you interested in Tissue Engineering to fix your child's ear? | 3.875 | 3.200 | 0.040 |
| Would you believe a doctor who says Tissue Engineering would help fix your child's ear? | 3.958 | 3.000 | 0.002 |

Significant differences in response rates to individual questions when groups are categorized by barrier differences including surveyors who indicated cost considerations, cultural considerations, and travel considerations for reconstruction. Averaged responses are indicated on a Likert scale of 0 (not at all) to 5 (definitely). Differences in response rates were deemed significantly different at p < 0.05.

### Hesitation differences *in* survey results

Surveyors who indicated they would choose TE over other methods were more familiar with TE prior to the introduction of educational material (2.50 vs. 1.50 and 2.50 vs. 1.50, $p < 0.05$ and $p < 0.05$, respectively) and indicated that cost played a role in their willingness to choose TE (3.80 vs. 2.50 $p < 0.01$) (Table 4).

### Survey domain differences

When assessing the relationship between variables and survey domains, there was a significant positive correlation between a doctor providing TE information and willingness to adopt TE reconstructive solutions ($p = 0.014$). An understanding of TE was significantly and positively correlated with a desire to pursue TE as a reconstructive treatment ($p < 0.001$).

### COI and insurance differences

Patients from low COI areas scored significantly lower regarding their understanding of how microtia/anotia occurs (OR=0.061, $p = 0.015$) and had a significantly higher odds of reporting that cost would play a significant role in choosing TE (OR=23.17, $p = 0.008$). A nonparametric analysis revealed that patients residing in areas with very high COI showed a greater willingness to explore and utilize TE treatment options ($p = 0.016$). Additionally, patients with public insurance scored significantly lower regarding whether they previously looked up information regarding "microtia" or "anotia" on their own (OR=0.178, 95% CI=0.03–0.93), $p = 0.049$) and scored significantly lower overall in their willingness to adopt TE solutions (OR=0.155, 95% CI=0.03–0.79), $p = 0.028$).

**Table 4. Hesitation/Trust Differences.**

| Question | Indicated Worries in TE (mean) | Denied Worries in TE (mean) | p-value |
|---|---|---|---|
| How do you feel about using Tissue Engineering for your child's ear reconstruction? | 4.750 | 3.920 | 0.037 |
| Are you interested in learning more about Tissue Engineering? | 4.500 | 3.920 | 0.035 |
| **Question** | **Would Choose TE (mean)** | **Would Not Choose TE (mean)** | **p-value** |
| Have you heard of 'Tissue Engineering' before today? | 2.500 | 1.500 | 0.034 |
| Did you know what 'Tissue Engineering' is before today? | 2.500 | 1.500 | 0.028 |
| Have you looked up information about 'Microtia' or 'Anotia' on your own? | 2.950 | 4.500 | 0.010 |
| Do you know the different ways doctors fix microtia/anotia? | 3.150 | 4.500 | 0.004 |
| Are you interested in Tissue Engineering to fix your child's ear? | 4.100 | 3.500 | 0.049 |
| Would the cost of Tissue Engineering for your child's ear be a big part of your decision to choose Tissue Engineering? | 3.800 | 2.500 | 0.008 |
| **Question** | **Indicated Trust in TE (mean)** | **Denied Trust in TE (mean)** | **p-value** |
| How do you feel about using Tissue Engineering for your child's ear reconstruction? | 4.129 | 3.000 | 0.036 |
| Are you interested in learning more about Tissue Engineering? | 4.065 | 3.000 | 0.018 |
| Are you interested in Tissue Engineering to fix your child's ear? | 3.903 | 2.500 | 0.002 |

Significant differences in response rates to individual questions when groups are categorized by hesitation/trust differences including surveyors who indicated worries for TE, willingness to choose TE, and trust in TE. Averaged responses are indicated on a Likert scale of 0 (not at all) to 5 (definitely). Differences in response rates were deemed significantly different at $p < 0.05$.

## Discussion

TE is a newer technology with ongoing development across various areas of healthcare. For microtia patients, TE holds promise as a viable option for ear reconstruction, making it essential to understand the perspectives of key stakeholders, including patients and their families [13]. This study explored the perspectives of microtia patients and their guardians regarding TE, further contributing to the growing body of literature on patient-centered approaches to ear reconstruction.

Our study highlights that TE for ear reconstruction is an unfamiliar concept among microtia patients and their families. This unfamiliarity likely stems from the advancing nature of TE, which is still in the process of being fully integrated into clinical practice [14]. Continued education is essential for patients and their families, as it has been shown to reduce pre-operative anxiety in pediatric plastic surgery significantly and may increase the likelihood of pursuing surgical intervention [15,16]. The findings presented underscore the need for structured educational initiatives to support patient awareness and the clinical translation of TE as it becomes realized.

This study identified discrepancies between patients and their guardians regarding their understanding of TE and how cultural influences may play a role in their decision to choose TE for reconstruction. While guardians traditionally serve as the primary clinical decision-makers, there is an increasing recognition of pediatric patients' desire for autonomy in surgical decision-making, particularly in plastic surgery [17]. Guardians who identified cultural factors as an influence in their willingness to pursue TE were less familiar with the concept of TE prior to educational intervention. Cultural perspectives and health literacy have both been shown to influence attitudes toward plastic surgery as well as surgical decision-making and outcomes [18,19]. The intersection of cultural values, health education, and health literacy may therefore play an important role in a guardian's or patient's receptiveness to TE as a reconstructive option for microtia.

White patients exhibited a greater understanding of TE prior to receiving educational materials. This reduced understanding of non-white patients might increase barriers to TE-based care [20,21]. Providers should consider these factors when discussing reconstruction options with families and connect them to appropriate resources to promote equitable care.

Cost considerations varied according to the severity of microtia. Guardians of patients with lower grade microtia reported that the cost of reconstruction had a greater influence on their decision-making. In comparison, patients with higher microtia grade, White and non-Hispanic patients, were less likely to consider cost as a barrier in choosing TE. Patients and guardians of children with less severe anomalies may be less inclined to invest in reconstruction, potentially due to a lower perceived psychosocial impact [22,23]. Given the increasing out-of-pocket expenses associated with plastic surgery, cost remains a critical barrier to care [24,25]. The cost of TE should be discussed with patients, and potential socioeconomic barriers to access should be further explored.

Families who demonstrated a better understanding of TE and those who expressed a general comfort with surgical interventions were more likely to indicate a willingness to travel to a hospital that offers TE for ear reconstruction. Given that TE is not universally available at all pediatric centers, these findings emphasize the importance of targeted outreach and education efforts. Outreach clinics and educational initiatives continue to expand access to pediatric plastic surgery services, which will be instrumental in increasing awareness and the availability of TE as a reconstructive option for patients with microtia [26]. Additionally, those who expressed initial hesitations regarding TE demonstrated a willingness to learn more about TE as a reconstructive option. Furthermore, these findings emphasize a positive correlation between a doctor providing TE information and willingness to adopt TE solutions. Patient trust and confidence in their plastic surgeon remain a primary determinant of overall satisfaction with surgical care [27]. Accordingly, plastic surgery clinics should prioritize efforts to enhance patient education through their providers and understanding of TE as a reconstructive option for microtia. Strengthening patient knowledge may foster trust in the procedure and the surgical team, ultimately supporting informed decision-making and increasing acceptance of TE, while also developing reconstructive technology.

Patients from underserved and/or underrepresented backgrounds experience disparities in delayed presentation and overall decreased access to pediatric surgical care [28–30]. Our study highlights this phenomenon, as patients residing in

lower COI areas showed trends of having a lower baseline understanding of how microtia/anotia occurs and significantly higher odds of cost influencing their decision to adopt TE. Alternatively, patients from higher COI areas were more willing to consider TE for microtia reconstruction. The trend in the data emphasizes the potential for discrepancies in TE surgical accessibility and inequity for patients of lower socioeconomic backgrounds seeking microtia reconstruction.

As TE based reconstructive care continues to roll out, implications for practice becomes important for providers. Patient interest in TE may grow with TE education, emphasizing the importance for patient-provider communication regarding the technology. Provider trust strongly influences willingness for patients to adopt TE, highlighting the role in clinician decision making and knowledge. Additionally, demographic disparities exist for patients seeking TE based reconstructive care. These findings suggest targeted educational approaches to increase awareness for diverse populations, policy and financial support to reduce inequities in access, and evaluation of long-term outcomes.

This study is limited by its single-institution design, which may restrict the generalizability of the findings across broader patient populations. Due to the small sample size within this study relative to the number of predictors, multivariate analyses may be subject to overfitting. Additionally, though standardized, variability in educational material may have introduced recall and information bias in considering retrospective assessment of TE familiarity. COI as a proxy for SES might also introduce a potential ecological fallacy as zip code level data may not be representative of the individual surveyor/patient in this study.

Future directions include validating the perspectives survey and expanding survey efforts through multi-institutional collaboration to validate the associations made in this study. Additionally, incorporating qualitative analyses to better understand the cultural, financial, and geographic barriers that may impact family interest in TE for microtia reconstruction.

## Conclusions

TE is a novel concept with limited familiarity among microtia patients and guardians. Our study suggests the importance of educational interventions in fostering understanding and addressing hesitations. Demographic factors, cost considerations, COI, and insurance type emerged as significant determinants in influencing TE consideration for microtia reconstruction. As TE continues to be integrated across healthcare fields, efforts to promote health education, address cultural and socioeconomic barriers, and promote shared decision-making between patients, families, and providers will be essential. This study contributes to a foundational, patient-centered framework for the development and application of TE that extends beyond microtia reconstruction.

## Supporting information

**S1 Fig. Supplemental Digital Content 1.** Educational material provided to patients/guardians.
(PNG)

**S2 Fig. Supplemental Digital Content 2.** Survey provided to parents/guardians.
(PDF)

**S3 Fig. Supplemental Digital Content 3.** Survey provided to 16+patient.
(PDF)

**S1 Data. De-Identified TE Perspectives Data 11.25.**
(XLSX)

## Author contributions

**Conceptualization:** Sierra Willens, Rahim Esfandyarpour, Miles J. Pfaff.
**Data curation:** Arya Sherafat, Asthon Rosenbloom, Lana Mamoun, Leo Alaniz.

**Formal analysis:** Arya Sherafat, Chizoba Mosieri, Asthon Rosenbloom, Lana Mamoun.

**Investigation:** Arya Sherafat, Chizoba Mosieri.

**Methodology:** Arya Sherafat.

**Project administration:** Arya Sherafat, James Antongiovanni, Chizoba Mosieri.

**Resources:** Sierra Willens, Miles J. Pfaff.

**Supervision:** Miles J. Pfaff.

**Validation:** James Antongiovanni, Miles J. Pfaff.

**Visualization:** Miles J. Pfaff.

**Writing – original draft:** Arya Sherafat.

**Writing – review & editing:** Arya Sherafat, James Antongiovanni, Chizoba Mosieri, Rahim Esfandyarpour, Mary Ziegler, Miles J. Pfaff.

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
