## [Decision Letter · Decision Letter 0]

21 Oct 2025

Dear Dr. Pfaff,

Thank you for submitting your manuscript to PLOS ONE. After careful consideration, we feel that it has merit but does not fully meet PLOS ONE’s publication criteria as it currently stands. Therefore, we invite you to submit a revised version of the manuscript that addresses the points raised during the review process.

We look forward to receiving your revised manuscript.

Kind regards,

Arash Ghalandarzadeh, Ph.D. 

Editor

PLOS ONE

Journal Requirements:

2. Please include captions for your Supporting Information files at the end of your manuscript, and update any in-text citations to match accordingly. Please see our Supporting Information guidelines for more information: http://journals.plos.org/plosone/s/supporting-information .

Reviewers' comments:

Reviewer's Responses to Questions

**Comments to the Author**

1. Is the manuscript technically sound, and do the data support the conclusions?

Reviewer #1: Partly

Reviewer #2: Yes

2. Has the statistical analysis been performed appropriately and rigorously?

Reviewer #1: Yes

Reviewer #2: Yes

3. Have the authors made all data underlying the findings in their manuscript fully available?

Reviewer #1: Yes

Reviewer #2: Yes

4. Is the manuscript presented in an intelligible fashion and written in standard English?

Reviewer #1: Yes

Reviewer #2: Yes

Reviewer #1: The study addresses an important and emerging area in reconstructive surgery—patient and guardian perspectives on tissue engineering for microtia—which has clear clinical and translational relevance. I commend the authors for undertaking this work, engaging directly with patients and families, and presenting novel data in a field where patient-centered perspectives are often underrepresented.

My comments below aim to strengthen the manuscript by clarifying the methods, ensuring appropriate use of statistical analyses, and refining the presentation of results and conclusions. I have organized my feedback into major and minor points, with the goal of providing constructive suggestions to enhance the rigor, clarity, and impact of this valuable contribution.

1. **Small Sample Size vs. Complex Multivariate Models**: The study involves 40 surveys and the use of multivariate/ordinal logistic regressions. Given the likely low events-per-variable ratio, there is a risk of overfitting. Please report the number of predictors per model, consider using simpler models, or apply penalized/Firth logistic regression with accompanying model diagnostics (AIC, goodness-of-fit). If none of these options are feasible, I recommend removing overambitious multivariate claims.

2. **Ambiguity Regarding Timing of Educational Material**: The methods state that an educational document was “distributed… prior to the initiation of the survey,” yet the results indicate participants’ familiarity was assessed “before receiving supplemental educational material.” This is contradictory; please clarify whether baseline familiarity was assessed before participants viewed the material (preferred) or retrospectively after viewing. If the latter, discuss potential recall and information bias.

3. **Insufficient Description of COI (Child Opportunity Index) Mapping and Categories**: Please describe how zip codes were mapped to the COI and detail the cut-offs used for “low,” “high,” and “very high.” Additionally, present the distribution of participants across COI strata, and discuss the potential ecological fallacy of using area-level COI as a proxy for individual socioeconomic status (SES).

4. **Abstract — Statistics**: Avoid stating specific p-values for numerous small comparisons in the abstract; instead, summarize the key statistically significant findings. Aim to keep the abstract concise.

5. **Definition of Abbreviations on First Use**: For instance, "COI" is used before its full name is explained in the Methods section. Ensure that all abbreviations are defined upon their first use for clarity.

6. **Consider Reporting Effect Sizes**: For key comparisons, include effect sizes (Cohen’s d, rank-biserial correlation) to better interpret clinical relevance beyond p-values.

7. **Add a Short “Implications for Practice / Next Steps” Paragraph**: Given the findings, briefly outline practical recommendations for clinicians (e.g., provider education about tissue engineering, targeted outreach to low-COI communities). However, tie these recommendations to the study limitations to ensure they remain proportional.

Reviewer #2: This manuscript is technically sounds and well-written. The reviewer only has few minor concerns.

1. Tissue engineering is a broad terminology. Upon presenting this TE idea to patients, how do authors be certain that their patients could have through understanding of TE in the application of ear reconstruction?

2. The authors should also consider if there is sex-difference factors in making the decision, especially the survey participants in this study are mostly female.

3. The table is very blur (the reviewer had hard time to go over each questions). It is suggested to make the visual presentation more appealing.

4. What are the other races in the survey participants?

**Do you want your identity to be public for this peer review?** For information about this choice, including consent withdrawal, please see our Privacy Policy

Reviewer #1: **Yes: ** Milad hosseini

Reviewer #2: No

---

## [Author Response · Author response to Decision Letter 1]

18 Nov 2025

The responses to comments have been included in a separate document attached and included here for reference:

1. **Small Sample Size vs. Complex Multivariate Models**: The study involves 40 surveys and the use of multivariate/ordinal logistic regressions. Given the likely low events-per-variable ratio, there is a risk of overfitting. Please report the number of predictors per model, consider using simpler models, or apply penalized/Firth logistic regression with accompanying model diagnostics (AIC, goodness-of-fit). If none of these options are feasible, I recommend removing overambitious multivariate claims.

a. We appreciate this feedback and agree with the reviewer on the risk of overfitting and have incorporated this feedback into the limitation portion of our study. Multiple analysis methods were included in this study with simple two-sample t-test as a mode of primary data results. Additional multivariate analysis were exploratory and aimed to provide insight through a greater context of the study. Multivariate claims, including patient demographic claims, were removed from this study to better reflect the primary data. Of note, this is single institution study with early data exploring patient perspectives. Disscussion on increasing sample size to validate the associations made in this study have been included in the future directions of this study.

2. **Ambiguity Regarding Timing of Educational Material**: The methods state that an educational document was “distributed… prior to the initiation of the survey,” yet the results indicate participants’ familiarity was assessed “before receiving supplemental educational material.” This is contradictory; please clarify whether baseline familiarity was assessed before participants viewed the material (preferred) or retrospectively after viewing. If the latter, discuss potential recall and information bias.

a. Thank you for this comment and feedback on our educational material distribution methods. Our survey included the following two questions to assess familiarity prior to distribution of the educational material: (1) “Have you heard of Tissue Engineering before today?” and (2) “Did you know what Tissue Engineering is before today?” Survey distribution was initiated after the introduction educational material and these two questions were used as retrospective proxys to assess familiarity prior to that material. For further clarification for the reader, the introductory sentence of our “Quantitative Survey Evaluation” portion of the methods in the manuscript has been adapted to reflect this. Additionally, a comment regarding potential recall and information bias has been added to the limitations of this study to provide additional context.

3. **Insufficient Description of COI (Child Opportunity Index) Mapping and Categories**: Please describe how zip codes were mapped to the COI and detail the cut-offs used for “low,” “high,” and “very high.” Additionally, present the distribution of participants across COI strata, and discuss the potential ecological fallacy of using area-level COI as a proxy for individual socioeconomic status (SES).

a. Thank you for this comment. Child Opportunity Index (COI) zip code mapping is done in accordance to the datasets available at diversitydatakids.org. As for the COI 3.0, which was used for our research, 44 content area indicators were incorporated into the score for COI including:

i. “Early childhood education, elementary education, secondary and post-secondary education, education resources, pollution, healthy environments, safety- and health-related resources, economic opportunities, economic resources, concentrated inequity, housing resources, social resources, wealth”

ii. Composite z scores of these indicators were calculated and grouped by the following:

iii. “census tracts with composite z-scores at or below the 20th population-weighted 2023 percentile were sorted into the “very low” group. Tracts above the 20th and at or below the 40th population-weighted 2023 percentile were classified as “low opportunity.” Tracts above the 40th and at or below the 60th population-weighted 2023 percentile were classified as “moderate opportunity,” tracts above the 60th and at or below the 80th population-weighted 2023 percentile were classified as “high opportunity” and tracts above the 80th population-weighted 2023 percentile were classified as “very high opportunity.””

b. These methods are further expanded upon in the manuscript in the “Data Analysis & Grouping” Section

c. The distribution of COI strata for our study is included in Table 1 which outlines the following: Very Low (5), Low (17), Moderate (5), High (3), Very High (10).

d. The ecological fallacy of using area level COI as a proxy for SES was added to the limitations portion of this manuscript.

4. **Abstract — Statistics**: Avoid stating specific p-values for numerous small comparisons in the abstract; instead, summarize the key statistically significant findings. Aim to keep the abstract concise.

a. The abstract has been adapted to exclude specific p values and contains only key statistical findings. The abstract has been made consice for clarity.

5. **Definition of Abbreviations on First Use**: For instance, "COI" is used before its full name is explained in the Methods section. Ensure that all abbreviations are defined upon their first use for clarity.

a. Thank you for this consideration. The manuscript has been screened for abbreviation definitions on first use and should be up to date. “Childhood Opportunity Index” is first introduced and defined as “COI” in on first use and is abbreviated in all of the following lines of the manuscript.

6. **Consider Reporting Effect Sizes**: For key comparisons, include effect sizes (Cohen’s d, rank-biserial correlation) to better interpret clinical relevance beyond p-values.

a. We appreciate the reviewers suggestion to include effect sizes however, given the exploratory nature of our study and smaller sample size, we chose not to include formal effect size estimates. We have, however, aimed to elaborate in the clinical relevance of our findings in the discussion.

7. **Add a Short “Implications for Practice / Next Steps” Paragraph**: Given the findings, briefly outline practical recommendations for clinicians (e.g., provider education about tissue engineering, targeted outreach to low-COI communities). However, tie these recommendations to the study limitations to ensure they remain proportional.

a. We agree with the reviewer that additional recommendations for clinicians should be added to the discussion of this manuscript. This has been added to the discussion portion of our manuscript. Our “implications for practice / next steps” paragraph aims to discuss the importance of provider education and targeted outreach for TE-based reconstruction.

---

## [Editor Report · Decision Letter 1]

20 Nov 2025

Patient and Guardian Perspectives on Tissue Engineering in Microtia Reconstruction

PONE-D-25-44361R1

Dear Dr. J. Pfaff,

We’re pleased to inform you that your manuscript has been judged scientifically suitable for publication and will be formally accepted for publication once it meets all outstanding technical requirements.

Kind regards,

Arash Ghalandarzadeh, Ph.D. Student

Editor

PLOS ONE

Additional Editor Comments (optional):

Reviewers' comments:

The author has adequately addressed all comments, and the manuscript is suitable for publication.

---

## [Editor Report · Acceptance letter]

PONE-D-25-44361R1

PLOS One

Dear Dr. Pfaff,

I'm pleased to inform you that your manuscript has been deemed suitable for publication in PLOS One. Congratulations! Your manuscript is now being handed over to our production team.

Kind regards,

on behalf of

Mr. Arash Ghalandarzadeh

Guest Editor

PLOS One